# Bioprospects of Endophytic Bacteria in Plant Growth Promotion and Ag-Nanoparticle Biosynthesis

**DOI:** 10.3390/plants11141787

**Published:** 2022-07-06

**Authors:** Monika Singh, Kamal A. Qureshi, Mariusz Jaremko, Minakshi Rajput, Sandeep Kumar Singh, Kapil D. Pandey, Luiz Fernando Romanholo Ferreira, Ajay Kumar

**Affiliations:** 1Department of Biotechnology, School of Applied and Life Sciences, Uttaranchal University, Dehradun 248007, India; monikasingh_bhu@yahoo.com (M.S.); m.rajput1991@gmail.com (M.R.); 2Centre of Advanced Study in Botany, Institute of Science, Banaras Hindu University, Varanasi 221005, India; kdpandey2005@gmail.com; 3Department of Pharmaceutics, Unaizah College of Pharmacy, Qassim University, Unaizah 51911, Saudi Arabia; ka.qurishe@qu.edu.sa; 4Smart-Health Initiative (SHI) and Red Sea Research Center (RSRC), Division of Biological and Environment Sciences and Engineering (BESE), King Abdullah University of Science and Technology (KAUST), Thuwal 23955-6900, Saudi Arabia; mariusz.jaremko@kaust.edu.sa; 5Division of Microbiology, Indian Agricultural Research Institute, Pusa, New Delhi 110012, India; sandeepksingh015@gmail.com; 6Department of Zoology, Pachhunga University College Campus, Mizoram University (a Central University), Aizawl 796001, India; kaushalpuc@gmail.com; 7Institute for Technology and Research (ITP), Tiradentes University-UNIT, Aracaju 49032-490, SE, Brazil; romanholobio@gmail.com

**Keywords:** endophytic bacteria, plant growth promotion, tomato, Ag-nanoparticles, HR-SEM, TEM, FTIR, XRD, anti-microbial activity

## Abstract

In this study, five endophytic bacterial strains, namely *Rhizobium pusense* (MS-1), *Bacillus cereus* MS-2, *Bacillus flexus* (MS-3), *Methylophilus flavus* (MS-4), and *Pseudomonas aeruginosa* (MS-5), were used to investigate their potential role in the enhancement of growth yields of two types of tomato varieties, viz. hybrid and local, and in the biosynthesis of silver nanoparticles (AgNPs). The inoculation of bacterial strains enhanced the root and shoot length, biomass, and leaf chlorophyll contents. The fruit weight of the tomato (kg/plant) was also higher in the bacteria inoculated plants of both hybrid and local varieties than in the control (untreated). A significant increase was recorded in the fruit yield (g/plant) in all the treatments, whereas *Methylophilus flavus* (MS-4) inoculated plants yielded nearly 2.5 times more fruit weight compared to the control in the hybrid variety and two times higher in the local variety. The response to *M. flavus* as a microbial inoculant was greater than to the other strains. Biosynthesis of Ag nanoparticles was also carried out using all five endophytic bacterial strains. The weakest producers of AgNPs were *Rhizobium pusense* (MS-1) and *Methylophilus flavus* (MS-4), while *Bacillus cereus* MS-2, *Bacillus flexus* (MS-3), and *Pseudomonas aeruginosa* (MS-5) were strong producers of AgNPs. Nanoparticles were further characterized using high-resolution scanning electron microscopy (HR-SEM), transmission electron microscopy (TEM), Fourier transform infrared (FTIR), UV-Vis spectrophotometry, and X-ray diffraction (XRD) analysis, and revealed cuboidal shaped AgNPs in the *Bacillus cereus* MS-2 strain. In addition, the biosynthesized AgNPs showed antibacterial activity against various pathogenic and endophytic bacterial strains.

## 1. Introduction

Producing a sufficient food supply for the rapidly increasing global population is a major challenge. With conventional agricultural practices, approximately 35% of crops become damaged during the cultivation period due to pest and microbial attacks, climatic disasters, and limited soil macro and micronutrient availability [1].

To obtain a fresh and healthy crop yield via organic farming, the most common and widely accepted approach among researchers and farmers is to use plant growth-promoting microorganisms (PGPMs), including bacteria, fungi, and actinomycetes, which modulate plant growth and also protect the host from pathogen invasion [2]. The common mechanisms of most PGPMs are phosphate solubilization, phytohormone modulation, synthesis of bioactive compounds, and increasing nutrient availability, etc. [3,4,5].

Endophytes are microorganisms that reside in the host without showing apparent signs of infection and play significant roles in plant growth even under stress conditions [6,7]. Recently, the bacterization of crops with suitable bacterial strains has been considered an excellent alternative to synthetic fertilizer. Microbes of plant origins, either epiphytes or endophytes, can be used as biological fertilizers, especially for horticultural fruits because of their raw consumption to avoid adverse effects. Endophytes have been reported in almost every part of the host plant and have the potential for better colonization efficacy, and are frequently employed as plant or soil inoculants to enhance growth [8,9].

The tomato is the second most important vegetable product after the potato, and is consumed in both cooked and uncooked forms, and is also transformed into industrial products (puree, canned, sauces) worldwide. In the recent past, various reports have been published regarding the tomato’s associated rhizosphere and endophytic microbiome and their potential applications in biotic and abiotic stress management [10,11]. In previous studies, several authors reported endophytic bacterial isolates of tomato strains [12]. Tian et al. [13] reported *Pseudomonas*, *Bacillus*, and *Rhizobium* sp. as endophytes, which are the core communities of tomato roots. Moreover, recently, various authors reported a significant enhancement in tomato yields after applying seed or soil microbial inoculants. The application of microbial inoculants significantly enhanced the morphological parameters and yields of bioactive compounds.

The synthesis of nanoparticles from growth-promoting microorganisms is a fascinating approach, and these nanoparticles have been frequently utilized for sustainable agriculture as nanofertilizers and nanopesticides [14,15]. Among the various biosynthesized nanoparticles, silver nanoparticles (Ag-nanoparticles) are one of the most important and efficient and are frequently used in various allied sectors. The formation of silver nanocrystallites in bacteria, yeasts, and fungi has been well documented in several published articles [16,17,18]. The biosynthesis of various metallic and oxide nanoparticles (NPs) using fungal strains *Fusarium oxysporum*, *Penicillium* sp. and some *Bacillus* strains have been frequently reported [19,20,21,22]. However, there are various factors, including pH, temperature, and silver nitrate concentration, which control the size of the biosynthesized AgNPs, as reported in previous studies [23].

In the present study, we used five endophytic bacterial strains and applied them to two varieties of tomato seeds and seedlings, to evaluate their impact on the morphological yields of tomatoes. Further, the presence of the bacteria in the endorhizosphere was confirmed by SEM analysis. In addition, the biosynthesis of silver nanoparticles was carried out using the endophytic bacterial strains and characterized by instrumentation techniques.

## 2. Materials and Methods

### 2.1. Sample Collection and Experimental Site

Two varieties of tomato S-22, a determinate hybrid variety (the first harvest starts from 55–60 days after transplanting) and another local variety S-3619 (the first harvest starts from 60 to 65 days after transplantation) were collected from the horticulture section of ICAR-I.I.V.R. Varanasi, India, and used in the experiments. Both varieties were shown to have optimum yields and higher amounts of bacterial populations, as reported in a previous study [24]. The experiments were carried out in the botanical garden of the B.H.U. campus, Varanasi (25° 20′ N and 83° 00′ E; elevation 80.71 m).

### 2.2. Inoculation of Endophytic Bacterial Strains on the Seeds and Seedlings of Tomatoes

Five endophytic bacterial strains viz. *Rhizobium pusense* MS-1 (Accession number: MF977751), *Bacillus cereus* MS-2 (Accession number: KY018604), *B. flexus* MS-3 Accession number: KX858538), *Methylophilus flavus* MS-4 (Accession number: MF977752) and *Pseudomonas aeruginosa* MS-5 (Accession number: KX817189, which were previously isolated from the roots of tomatoes, were stored in the Laboratory of Environmental Microbiology, Department of Botany, B.H.U. [25]. The bacterial strains were grown to the mid-log phase, and an adjusted cell biomass (~10^9^ cells ml^−1^) was harvested using centrifugation (4000 rpm, 5 min at 4 °C), and suspended in phosphate buffer saline (PBS, pH-7.5 10^−2^ M). To determine the density and purity of the microbial culture for inoculation, an adequate amount of each strain culture was serially diluted in PBS and plated on nutrient agar media, then 10^8^ CFU/gm was used to inoculate the tomato seeds, which were then stored at room temperature for 24 h. Then the inoculated seeds were planted in polybags and earthen pots in the botanical garden of the B.H.U. campus. The inoculated seeds and seedlings were grown in the pots (soil: sand: compost: vermicompost; 2:2:1 ratio), under natural conditions with sufficient soil moisture. Necessary care was taken during the developmental growth of the tomato plants.

### 2.3. Preparation of Inoculants and Slurry

Endophytic bacterial strains (10^8^ cells per gram) were mixed thoroughly in moss peat soil (pH 6.8) and stored in plastic trays for one month under laboratory conditions. Then, further inoculants were dried in the laboratory in the absence of sunlight and mixed to make a fine powder. The moss peat soil-based inoculants of different entophytic bacteria were used for the preparation of the respective slurry.

Surface sterilized seeds were impregnated with the bacterial gum-Arabic suspension (Jiggery 10 gm was dissolved in boiled (100 °C) warm water and mixed with gum Arabic (40% *w*/*v*) for 5 min and then dried on filter paper after pelleted with calcium carbonate powder (mesh size 300) for 24 h. Finally, the bacterized seeds were transferred to polybags and earthen pots containing sterilized soil, sand, and compost (2:1:1) [26].

### 2.4. Seed and Seedling Bacterization and Biometry

Seeds of hybrid and local tomato varieties were surface sterilized using 2% NaOCl (sodium hypochlorite) for two min followed by washing with sterile double distilled water (3 times) and drying on blotting paper to absorb excess moisture. Seeds were inoculated with inoculants slurry and pelleted with fine calcium carbonate powder. The treated seeds were sown in earthen pots and polybags (soil, sand, and compost: 2:1:1 ratio) and allowed to germinate. Triplicates were used for each treatment and the data of germination were recorded after every 2 days. Seeds without treatments were used as the control. After 21 days of sowing, the seedlings were uprooted from the earthen pots and the leaves of the plants were counted manually and the height was measured. Shoot and root dry biomass were calculated after drying in an oven at 60 °C [27]. The uprooted seedlings were washed with tap water and again bacterized with the endophytic bacterial strain slurry for 30 min. The inoculated seedlings were transplanted in earthen pots containing soil as described above.

### 2.5. Internal Root Colonization by Bacteria (Scanning Electron Microscopy or SEM)

For SEM analysis of bacterial inoculated and control plants (40 days old), the roots of two tomato varieties were collected from the botanical garden and cut into thin transverse sections using an ultramicrotome. The root samples were cleaned 2–3 times using sterilized distilled water just after removing the root-adhering soil and transferred into small screw-capped tubes. The transverse sections were fixed in 2.5% (*v*/*v*) glutaraldehyde for 1 h in 0.1 M PBS and then further in 1% (*w*/*v*) osmium tetraoxide (OsO_4_) for 40 min. The fixed samples were dehydrated using ethanol (20−100%; (*v*/*v*) for 2–3 min in each concentration) [28], and then dried in a critical point dryer, and visualized using a Scanning Electron Microscope with GEMINI column (ZEISS EVO18 Research, Carl Zeiss Microscopy GmbH, Jena, Germany).

### 2.6. Estimation of Chlorophyll Content in Leaves

One gram of finely cut fresh leaves of 40-day-old plants was taken for estimation, and leaves were ground with 20–40 mL of 80% acetone. The leaf pieces dipped in the acetone were incubated in the dark for 24 h and then centrifuged at 10,000× *g* rpm for 5 min. The supernatants were collected and re-centrifuged to clean the residue. The clean samples were subjected to optical density (O.D.) measurements at a wavelength of 645 nm and 663 nm in a UV-vis spectrophotometer using solvent acetone as blank [29].

### 2.7. Synthesis of Nanoparticles

The endophytic bacterial strains were cultured in sterile Luria, King’s B, Jenson, and NMS broth and incubated in a shaking incubator (150 rpm per min) for 72 h at 45 ± 1 °C. Cell pellets were harvested by centrifugation at 9000 rpm for 12 min. The pellets were washed using sterile double distilled water to clean or remove extra nutrients from the media harvested cell biomass. Then 1 g cell biomass was re-suspended in 20 mL of 1 mM AgNO_3_ and stored at 45 ± 1 °C for 72–120 h [30].

#### 2.7.1. Nanoparticle Characterization

UV-Visible Spectrophotometry

Ag-nanoparticle synthesis was primarily screened by observing changes in the color of the reaction mixture from orange to brown. Synthesis was further confirmed by detecting surface plasmon resonance (SPR) of AgNPs in the UV–Vis spectrum (200–700 nm) at a resolution of 1 nm using a spectrophotometer (U-2600, Hitachi). The λ max is reported to be 415–420 nm for silver nanoparticles. During the experiment, absorption spectra were recorded every 24 h until no further significant increase in the peak was observed [30].

Fourier Transform Infrared (FTIR) Spectroscopy Observations

For FTIR spectroscopy, the reduced AgNO_3_ solution was filtered using Millipore filters, then centrifuged at 5000× *g* rpm for 20 min and lyophilized. Next, the lyophilized samples were ground using KBr pellets for further use [31]. However, to check biomolecular capping to the surface of AgNPs, FTIR spectroscopy was performed using a JASCO 6700 FTIR spectrophotometer with a resolution of 4 cm^−1^ and a range of 400–4000 cm^−1^.

X-ray Diffraction Analysis

The lyophilized samples of the biosynthesized silver nanoparticles coated on XRD grids were studied for X-ray diffraction patterns using the Philips PAN analytical X’ Pert XRD System operated at a voltage of 30 kV and current of 20 mA with a scan rate of 0.03 0/s using CUKα radiation. The presence of bacterial synthesized Ag-nanoparticles in different phases was determined using X′ pert high score software.

High-Resolution Scanning Electron Microscopy (HR-SEM)

For HR-SEM analysis, the magnification was set in the order of 100,000× and resolution of 0.4 nm. The aqueous solution of synthesized silver nanoparticles was lyophilized to obtain a pure powdered form of AgNPs. The powdered particles were dissolved in acetone at a concentration of 1 mg/mL. The Ag nanoparticle solution was air-dried on the specimen grid and observed with a SUPRA40, Zeiss HR-SEM (Germany).

Transmission Electron Microscopy (TEM)

The sizes of the biosynthesized AgNPs were determined by TEM (TECNAI G2-TWIN-FEI TEM, The Netherlands). Samples were prepared for photography using ultrasonication of biosynthesized AgNPs, and a drop of the acetonic sample was placed on the carbon-coated copper grid and allowed to dry overnight under vacuum conditions. TEM was carried out under an accelerating voltage of 200 kV. The histograms of the AgNPs particle size distribution were prepared using Image J software. For the histogram preparation, the total number of NPs counted at 50 nm in the area of 15 × 20 cm^2^, were 98, 176, and 89 for *B. flexus* MS-2, *B. cereus* MS-3 and *P. aeruginosa* MS-5, respectively.

#### 2.7.2. Antibacterial Activity

The agar well diffusion method was followed to evaluate the antibacterial activity of the Ag-nanoparticles [32] against different bacterial strains viz. *Bacillus thuringiensis, Azotobacter chroococcum* (CL13), *Escherichia coli*, *Pseudomonas putida* (ECL5), *Bacillus licheniformis* (R-1), and *Rhizobium* sp. (CV1); 1 mL of a 24 h old bacterial strain was spread uniformly on the nutrient agar plates, and further wells were prepared on the culture plates and 100 μL of 1 mM biosynthesized AgNP solution was loaded and incubated at 38 ± 1 °C for 48–72 h. The clear halo zones around the wells were measured to evaluate antibacterial activity.

### 2.8. Statistical Analysis

The experimental data were analyzed using one-way and two-way analysis of Variance (ANOVA) using SPSS 16.0 software. All the experiments were performed in triplicate for accuracy, and values in figures and tables represent the arithmetic mean of the three replicates. All results presented are mean values of three replicates, and the total particle sizes of the synthesized silver nanoparticles from the TEM images were calculated using Image J software.

## 3. Results

For these experiments, five endophytic bacterial strains, namely Rhizobium pusense (MS-1), Bacillus flexus (MS-2), B. cereus (MS-3), Methylophilus flavus (MS-4), and Pseudomonas aeruginosa (MS-5) were inoculated in two different varieties of the tomato and grown in polybags and earthen pots. Polybags and earthen pots were used for growing seedlings while earthen pots were selected for the better growth and yield of the tomatoes (Figure 1). All the strains showed positive results compared with the control.

### 3.1. Effect of Endophytic Bacterial Inoculation on the Growth of Tomato Seedlings

The fresh weights (g) of 21-day-old seedlings were higher in the bacterial inoculated seeds compared to the control. *R. pusense* (MS-1) increased fresh shoot and root wt. of the local variety of tomato seedling by 50.76% and 41.37% respectively, in comparison with the control, while in the hybrid, 12.02% incensement in fresh shoot wt. and 55.17% in fresh root wt. was recorded. *Bacillus flexus* (MS-2) also produced positive effects on the biomass of roots and shoots, by 68.29% shoot and 39% fresh root wt. in the local variety, while 15.24% fresh shoot and a nearly 2 times root biomass increase was observed in the hybrid variety compared to the controls. *B. cereus* (MS-3) also showed positive results on the root, and shoot biomass of the bacterial inoculated plant by increasing 35.74% fresh shoot and 82% fresh root wt. in the local variety, and 4.22% and 98% enhancement in the hybrid variety shoot and root biomass, respectively. *Methylophilus flavus* (MS-4) had strong effects on the root growth in local and hybrid varieties. In the local variety, fresh shoot biomass increased only 41% while the root weight increased 2.5 times compared with the control. Hybrid variety inoculated with bacterial isolate MS-4 showed 86% enhancement in the fresh shoot and 2.25 times in the fresh root weight, compared to the control. *P. aeruginosa* (MS-5) showed 87% increase in fresh shoot wt. and a nearly two-time increase in fresh root wt. of the local variety, while 25.30% and a two-time increase in fresh shoot and root wt. respectively, compared with the control. All data are presented in Figure 2.

### 3.2. Growth Parameters and Yields of Tomato after Endophytic Bacterial Inoculation

The root and shoot length, wet and dry biomass, and leaf number of bacterial inoculated plants were significantly (Multiple comparison, Tukey HSD, significant at *p* ≤ 0.5) increased in both varieties compared with the control. Inoculation with *M. flavus* (MS-4) in the hybrid variety showed the maximum increase in shoot and root length by 73.3% and 51.2%, respectively. This strain also increased the maximum shoot and root length in the local variety by 83% and 68% respectively (Table 1).

All the agronomical traits data showed significant increases (Multiple comparison, Tukey HSD, significant at *p* ≤ 0.5). However, in both the varieties, either local or hybrid, the days of germination in the endophytic inoculated seeds were found to be greater than or equal to the control (un-inoculated). The percentage germination of the local variety was higher than the hybrid variety. The maximum germination percentage of hybrid and local varieties was recorded in *M. flavus* (MS-4) inoculated plants compared to other treatments and the control.

The bacterization of seeds and seedlings significantly improved the height, number of fruits, and fruit weight of both the varieties of tomato (Table 2). The number of fruits of hybrid variety tomato plants increased by 5.2% with *R. pusense* (MS-1), 15.7% with *B. cereus* (MS-3), 26.3% with *B. flexus* (MS-2), 31% with *P. aeruginosa* (MS-5) and the maximum of 63.5% with *M. flavus* (MS-4) compared with un-inoculated control. In the local variety, the percent increase in the number of fruits was more than in the hybrid variety. The bacterial inoculated plants of the local variety showed 13% increase in the number of fruits by *R. pusense* (MS-1), 30.0% by *B. cereus* (MS-3), 39% by *B. flexus* (MS-2), 52% by *P. aeruginosa* (MS-5) and the maximum percent of 65% by *M. flavus* (MS-4) compared with the control. The fruit weight also increased in bacterial inoculated tomato plants of both varieties. In the hybrid variety, MS-4 showed a maximum 68% increase followed by 60% with *P. aeruginosa* (MS-5), 40% with *B. flavus* (MS-2), 20% with *B. cereus* (MS-3), and only 8% with *R. pusense* (MS-4), while in the local variety *M. flavus* (MS-4) also showed the maximum 81% increase, followed by *P. aeruginosa* (63%), *B. flavus* (54%, MS-2), *B. cereus* (40%, MS-3) and *R. pusense* (36%, MS-1) compared to their respective controls. The increase in height of plants of the hybrid variety was in the order of Control (C) < *R. pusense* (MS-1)< *B. cereus* (MS-3) < *B. flexus* (MS-2) < *P. aeruginosa* (MS-5) < *M. flavus* (MS-4), and the same for the increase in plant height of the local variety: Control (C) < *R. pusense* (MS-1) < *B. cereus* (MS-3) < *B. flexus* (MS-2) < *P. aeruginosa* (MS-5)< *M. flavus* (MS-4) inoculated tomato plants.

### 3.3. Estimation of Chlorophyll Content in Endophyte Inoculated Tomato Leaves

The mature (45 days old) leaves of bacterized tomato plants and controls were used for chlorophyll content estimation. Chlorophyll *a* and chlorophyll *b* were estimated in both varieties (hybrid and local) of tomato plant leaves. In both the varieties of tomato plants, the chlorophyll *a* content was more than chlorophyll *b*. Statistical analysis revealed significantly higher chlorophyll *b* in both the varieties of control as well as bacterial inoculated plant leaves (Figure 3).

In the local variety, chlorophyll *a* content increased 5% by MS-1, 9.2% by MS-2%, 8.6 % by MS-3, 35% by MS-4, and 28% by MS-5, while chlorophyll b content increased 3.5% by MS-1, 8.3% by MS-2, 10.8 by MS-3, 26.5% by MS-4 and 23.1% by MS-5, when compared with the control. In the hybrid variety, chlorophyll *a* content increased 2.5% by MS-1, 25% by MS-2%, 18% by MS-3, 36% by MS-4, and 33% by MS-5 while chlorophyll *b* content increased 3% by MS-1,18% by MS-2, 13.2 by MS-3, 30.5% by MS-4 and 26.1% by MS-5 compared to the control.

### 3.4. Root Colonization of Endophytic Bacterial Strains

The colonization efficacy of microbial strains is an important characteristic of endophytes and strains with higher colonization efficacy are selected as plant and soil inoculants. Seedling (40 days old) endophytic bacteria inoculated and uninoculated control plants of the hybrid variety were taken, and transverse sections of the roots were visualized by scanning electron microscopy (SEM). The maximum number of the bacterized strains were seen in the root cortex region followed by xylem vessels in the endophyte inoculated tomato plants at 10.00 KX magnification. In contrast, bacterial density in the uninoculated control was low and the population of bacteria was scattered (Figure 4).

### 3.5. Biosynthesis and characterization of nanoparticles

All five strains were evaluated for their ability to mediate the biosynthesis of silver nanoparticles. Excellent growth was observed after 24 h in all the bacterial strains *Viz. Rhizobium pusense* (MS-1,) *B. flexus* (MS-2), *B. cereus* (MS-3), *Methylophilus flavus* (MS-4) and *P. aeruginosa* (MS-5) on respective media for enhanced growth.

The initial characterization and formation of Ag-nanoparticles were screened by observing changing color. The changes in color from pale white to brown indicate the formation of nanoparticles. This was further confirmed by the reduction of AgNO_3_ to AgNPs, which produces a characteristic sharp peak at 415–420 nm wavelength in the UV-visible spectra. However, out of five endophytic bacteria, only three strains (MS-2, MS-3, MS-5) were evaluated for the biosynthesis of AgNPs, because the characteristic peaks corresponding to AgNPs were observed only in these three strains. Although the silver nitrate solution (1 mM) was treated with MS-2, MS-3 and MS-5 strains at different time intervals (6, 12, 18, 24, 48 and 72 h), the maximum synthesis was recorded after 72 h (Figure 5).

FTIR spectroscopy revealed the functional groups of the biomolecules responsible for the capping and stabilizing of the silver nanoparticles. The FTIR spectral analysis of AgNPs synthesized by *B. flexus* (MS-2) showed intensive peaks at 3445, 3268, 2923, 1773, 1623, 1361, and 1035 cm^−1^. *B. cereus* (MS-3) exhibited intensive peaks at 3799, 3452, 2924, 1736, 1628, 1361, and 1213 cm^−1^, whereas the aqueous extract of *P. aeruginosa* (MS-5) showed only two intense peaks at 3274 and 1635 (Figure 6). The crystalline nature of the synthesized AgNPs was established by X-ray diffraction analysis at 2 θ values, ranging from 27 to 85, showing eight intense peaks.

The comparison of the XRD patterns of our data with the standard [33] confirmed the formation of AgNPs nanocrystals, as evidenced by the peaks at 2 θ values of 27.8°, 32.6°, 38.8°, 46.3°, 54.9°, 57.8°, 67.7° 76.8° and 85.7° corresponding to (111), (200), (220), (311), (222), (400), (331), (420), and (422) planes.

However, the size of the AgNPs ranged from 10 nm to 90 nm synthesized at the temperature 45 ± 1 °C. The shapes of the AgNPs synthesized by *B. flexus* (MS-2) and *P. aeruginosa* (MS-5) were spherical, while *B. cereus* (MS-3) synthesized AgNPs were both spherical and cuboid, observed by HR-SEM (Figure 7).

TEM analysis showed well-dispersed AgNPs and sizes were present in the range of 3–39 nm (Figure 8). However, the morphologies of the synthesized nanoparticles were observed mostly in the spherical and cubical form. In detail, the AgNPs synthesized by *B. flexus* (MS-2) were in spherical form with 3–28 nm range; *B. cereus* (MS-3) synthesized AgNPs were also in spherical and cuboid forms with 3–26 nm range. However, *P. aeruginosa* (MS-5) was spherical in shape with a maximum size range of 18–39 nm.

### 3.6. Antibacterial Activity

The AgNPs biosynthesized by *B. flexus* (MS-2), *B. cereus* (MS-3), and *P. aeruginosa* (MS-5) showed significant antibacterial activities against pathogenic *E. coli* and some endophytes.

The results demonstrate that AgNPs biosynthesized by *P. aeruginosa* (MS-5) showed a maximum zone of inhibition against *B. thuringiensis* (21 ± 2), *P. putida* (28 ± 2), *A. chroococcum* (9 ± 0.8), and *Rhizobium* sp. (11 ± 1), while AgNPs biosynthesized by *B. flexus* (MS-2) and *B. cereus* (MS-3) showed a maximum zone of inhibition against *E. coli* (7 ± 0.6) and *B. licheniformis* (10 ± 0.8), respectively, whereas no zone of inhibition was recorded in the controls (AgNO_3_ solution) (Table 3, Figure 9).

## 4. Discussion

In the last two decades, significant progress in microbiome research has been made especially in the field of sustainable agriculture as biofertilizers and mitigation of biotic and abiotic stress [34]. Generally, large populations still rely on chemical fertilizers or pesticides to meet the requirement of nutrients in the soil and control phytopathogens during pre or postharvest conditions [34]. However, today, the utilization of plant growth-promoting microorganisms, either epiphytes or endophytes, are a preferred alternative to agrochemicals in sustainable agriculture [35,36,37,38].

In our study, all five endophytic strains related to the genus *Bacillus*, *Pseudomonas*, *Rhizobium*, and *Methylophilus*, had significant growth promotion effects on plant development and also showed positive effects on chlorophyll content. Seeds and seedlings inoculated with endophytic bacterial strains were found to have stimulatory effects on root and shoot length fresh biomass, in both the varieties. In the latest published review, Omomowo and Babalola [39] briefly discussed the bacterial and fungal endophytes used for sustainable agricultural production. In previous studies, authors reported beneficial effects on plant growth after inoculation with plant growth-promoting microorganisms. For instance, inoculation of *Pseudomonas tolaasii* IEXb, and *Pseudomonas koreensis* SP28 have stimulatory effects on the plant height and shoot weight of *Zea mays* L. [40]. Kumar et al. [41] reported that different endophytic strains of *Bacillus* and *Pseudomonas* were positive in plant growth-promoting trait analyses. Fang et al. [42] reported that the strain *Pseudomonas aurantiaca* JD37 had excellent colonization potential and also promoted maize growth. *Methylobacterium* strains were also identified as having effects on seed germination, phytohormone modulation [43,44,45], nitrogen fixation [46,47], and also in protection from pathogen invasion [48,49]. Our study also reported similar observations after seed treatment with *Methylobacterium* in tomato varieties. Another genus *Pseudomonas* with several strains has shown excellent plant growth-promoting and biocontrol activities and is frequently employed for sustainable agriculture [50,51].

Both the varieties of tomato plants inoculated with PGPB endophytic bacteria strains showed a greater increase in chlorophyll *a* than chlorophyll *b* content. Chlorophyll *b* participates in the formation of light-harvesting complexes, whereas Chlorophyll *a* plays a key role in the formation of reaction centers. PGPR inoculation increases the physiological properties of a plant’s chlorophyll and photosynthetic rates [52]. All the endophytic bacterial strains inoculated in tomato varieties showed positive results in terms of the chlorophyll *a* and *b* content of the leaves. In a previously published report, the PGPR strains, alone or in combination, increased the rate of photosynthesis, chlorophyll content, and transpiration [53]. SEM has been used effectively to observe bacterial populations on root surfaces and endorhizospheres. This technique is useful for detecting microbial colonization in the different locations of the host tissue. SEM studies in tomato roots also revealed the characteristics of colony patterning and their morphology after inoculation with bacterial isolates compared with control. SEM of 40-day-old plant roots was performed to visualize the microbial colonies at 10.00 K X magnification and revealed a strong capacity to colonize the root cortex intercellular and intracellular spaces. A similar colonization pattern was also reported in SEM studies of *Burkholderia cepacia* strain Lu10-1 when inoculated with *Morusalba* L. and *P. Fluorescens* WCS365 in tomatoes [54,55]. Bacilio-Jimenez et al. [56] also evaluated endophytic strains in rice seeds through SEM analysis. PGPB has the potential to improve yield factors without any change in fruit quality [57]. All the endophytic bacterial strains inoculated in tomatoes increased the number of fruits, fruit weight and size, and finally the total yield. In a previous study, *Pseudomonas* BA-8 and *Bacillus* OSU-142 strains enlarged the fruits of sweet cherry plants after foliar application [58]. In another study, *Bacillus* was used to enhance the growth and suppression of wilt disease in tomatoes [59]. Similarly, a significant enhancement in the growth and fruit yield of tomato was observed after inoculation with *Pseudomonas fluorescens* [60].

The biosynthesis of metal nanoparticles from biological tools is now gaining momentum in the last two decades, due to their multifarious and potential applications in eco-friendly and efficient novel technologies of sustainable agriculture and healthcare management [61]. In a recently published review, Meena et al. [62] briefly discuss the synthesis and application prospects of nanoparticles synthesized by endophytic microorganisms.

Silver is one of the most widespread metals applied in various allied sectors [63]. In another review, Gerg et al. [64] briefly summarize Ag-nanoparticles synthesized by biological routes and their application in various sectors. The range of sizes of nanoparticles reported is 1–100 nm, and observed shapes are spherical, triangular and hexagonal. The shape of AgNPs is also found to be spherical and cuboidal in nature. In our study, the HR-SEM images of AgNPs synthesized using endophytic bacterial cell pellets were spherical and cuboid in shape with diameters ranging from 18 nm to 90 nm. TEM images of silver nanoparticles also confirmed their cubical and spherical morphology.

In this study, the development of silver nanoparticles was observed by the change in the color of the solution. The UV-Vis spectroscopy study revealed that AgNPs synthesized by *B. flexus* (MS-2) and *B. cereus* (MS-3) had nearly similar maximum absorbance (λ peak) from that of *P. aeruginosa* (MS-5) synthesized AgNPs. The results showed differential synthesis patterns of endophytic bacterial synthesized nanoparticles. In the reaction tubes, the bacterial biomass reduces the AgNO_3_ into AgNPs, which means that the spectroscopic behavior of AgNPs synthesized by different bacterial strains may differ [65].

The identification of functional groups present in the bacterial cell pellet was carried out using FTIR analysis. The sharp peak *B. flexus* (MS-2) synthesized AgNPs at 3445 cm^−1^ can be assigned to the N–H stretching of the primary amines; however, the peak at 2923 cm^−1^ corresponds to C–H or methylene groups of the protein [66]. The peak at 1623 cm^−1^ corresponds to asymmetrical C–O stretching [67], whereas 1361 cm^−1^ corresponds to the C–N groups of aromatic and aliphatic amines [68]; The peak at 1035 cm^−1^ showed O–H deformation/C–O stretch of the phenolic and alcoholic groups [69]. In our study, intense peaks were observed at 1635 and 3264 cm^−1^ in the FTIR spectra of *P. aeruginosa* (MS-5). Similar results of AgNP synthesis using *Bacillus cereus* have been reported by Sunkar et al. [70].

The spectrum comparison of X-ray diffraction in the study with the standard sample confirmed Ag nanoparticles were in the form of nanocrystals. Similar peaks were also found in the recently published laccase Ag/AgCl nanoparticles [19,71] and AgNPs by *Fusarium oxysporum* [72]. Biosynthesized silver nanostructure synthesis by *Gelidiella acerosa* extract also showed nearly a similar XRD pattern at 28.9°, 32.9°, 46.4°, 56°, and 58° [73]. The aggregation of AgNPs synthesized by *P. aeruginosa* (MS-5) could be associated with the drying process of the sample preparation in the TEM analysis. The AgNPs synthesized by the strains MS-2, MS-3 and MS-5 were different in size-range and their frequency differed from bacterial strains. A similar result was also reported for AgNP synthesis using *E. coli* [17].

Antibacterial analyses of Ag-nanoparticles revealed antibacterial activity against pathogenic and some endophytic bacterial strains. Similar AgNPs synthesized by *Phyllanthus amarus* and *Tinospora cordifolia* also showed efficient antimicrobial efficacy [74]. In our study, the Ag-nanoparticles synthesized by all three bacterial strains showed some degree of antibacterial activity. AgNPs biosynthesized by *P. aeruginosa* (MS-5) showed a maximum zone of inhibition against *B. thuringiensis*, *P. putida*, *A. chroococcum*, and *Rhizobium* sp. while AgNPs biosynthesized by *B. flexus* (MS-2) and *B. cereus* (MS-3) showed maximum zones of inhibition against *E. coli* and *B. licheniformis*.

In a previous study, it was reported that metallic nanomaterials showed the highest bactericidal activity due to their large surface-to-volume ratios and crystallographic surface structure [75]. However, the bactericidal activity of the Ag-nanoparticles also depends upon the present charge ion and shape. Abbaszadegan et al. [76] reported that positive-charged Ag nanoparticles have the highest antibacterial activity against all tested pathogens namely *Staphylococcus aureus*, *Streptococcus mutans*, and *Streptococcus pyogenes*, *E. coli* and *Proteus vulgaris*, while neutral nanoparticles have an intermediate effect, and negatively charged nanoparticles have the least activity against all the tested microorganisms. Pal et al. [77] reported that the shape of the Ag nanoparticles also plays a significant role in bactericidal property. Truncated triangular Ag-nanoparticles with {111} lattice plane showed strong antibacterial activity against *E. coli* compared with spherical and rod-shaped nanoparticles. Our study showed differential responses of the endophytic bacterial strains in Ag-nanoparticle synthesis, which may be strain-specific and depend on their different sizes and shapes.

## 5. Conclusions

In the present scenario of uncertainty of food security and the rising global population, there is an urgent need to find alternative resources that can significantly enhance agricultural yields and efficiently reach the target sites. In this study, we used five endophytic bacterial strains previously isolated from the roots of tomatoes, and seed and seedling treatments significantly enhanced the root and shoot length and biomass. In addition, they also significantly impacted the chlorophyll content of both local and hybrid varieties of tomatoes. Notably, the performance of *Methylophilus flavus* was superior in yield enhancement compared to the other bacterial strains, *Rhizobium pusense*, *Bacillus cereus*, *B. flexus*, and *Pseudomonas aeruginosa*. The utilization of nanoparticles in sustainable agriculture is a fascinating approach. In our study, *Bacillus* sp. and *Pseudomonas* sp. significantly promoted the synthesis of Ag-nanoparticles with potential antibacterial activities. These strains could be employed in high-temperature regions for sustainable agriculture, and strains, such as *Bacillus* and *Pseudomonas* can also be used for the synthesis of various bioactive nanoparticles.

## Figures and Tables

**Figure 1 plants-11-01787-f001:**
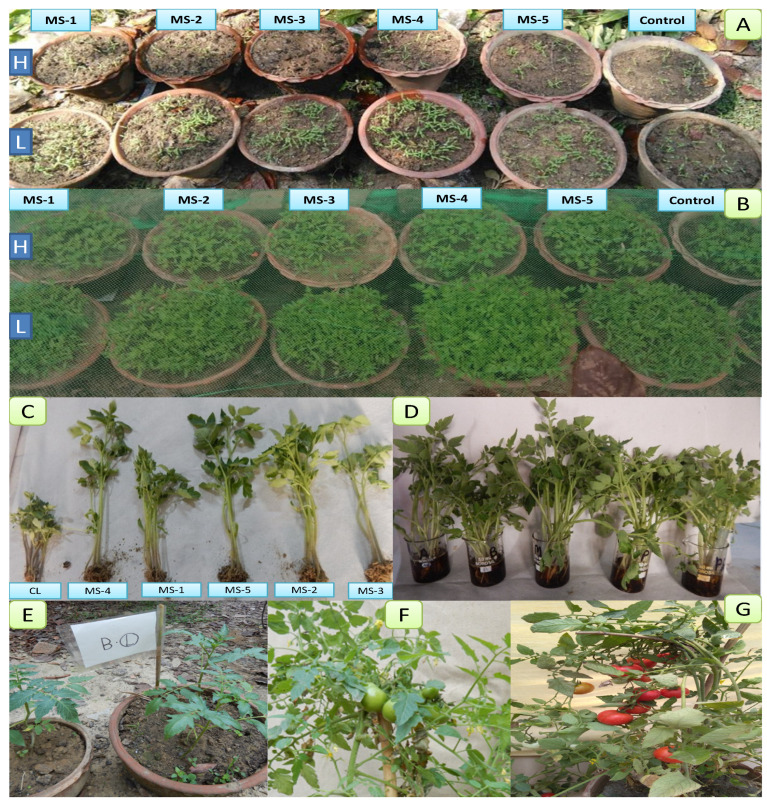
Different growth stages of tomato after bacterization of seeds and seedlings: H: hybrid, L: local; (**A**) sowing of bacterized seeds and control seeds germination (100 seeds per entity); (**B**) growth of tomato seedlings (10 days) under the net; (**C**) 21-day-old, uprooted seedlings; (**D**) bacterization of seedlings (21 days old); (**E**) seedlings transplanted into earthen pots after bacterization; (**F**) green fruits; (**G**) mature flower and fruits.

**Figure 2 plants-11-01787-f002:**
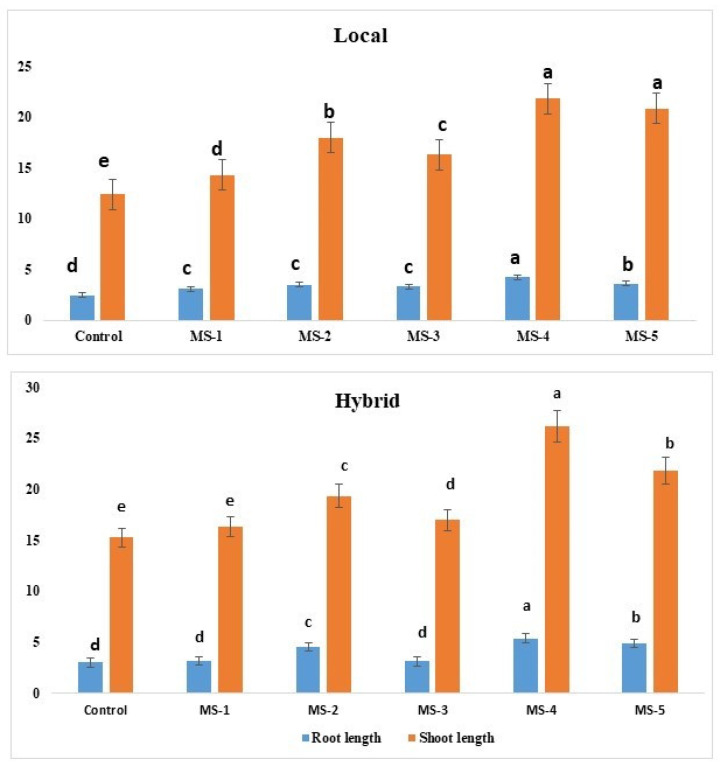
Effect of endophytic bacteria inoculation on the fresh root and shoot weight of tomato seedlings of local and hybrid varieties: Bars are the mean ± SD of three replicates. Bars labeled with different letters (a–e) are statistically significant at *p* < 0.05 (Tukey’s post hoc test).

**Figure 3 plants-11-01787-f003:**
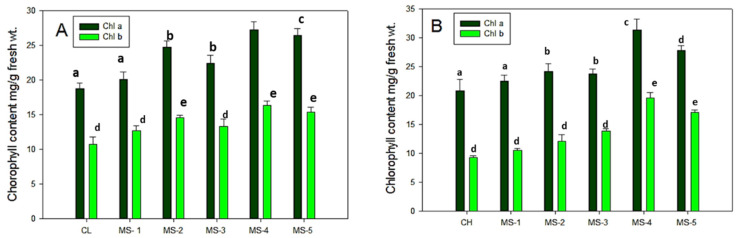
Chlorophyll content in leaves of endophytic inoculated tomato plants l. (**A**) Local; CL-Control local. (**B**) Hybrid variety, CH- Control hybrid; Data are the mean of triplicate datasets and values labeled with different letters (a–e) are statistically significant at *p* < 0.05.

**Figure 4 plants-11-01787-f004:**
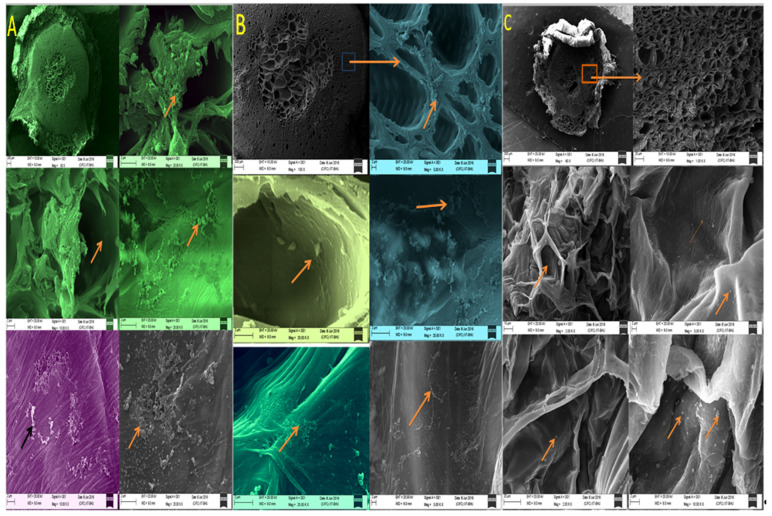
SEM images: Colonization of PGP endophytic bacteria in the roots of 40-day-old tomato seedlings, inoculated with bacterial strains. (**A**) *Methylophilus flavus* MS-4, (**B**) *P. aeruginosa* (MS-5), and (**C**) control (un-inoculated).

**Figure 5 plants-11-01787-f005:**
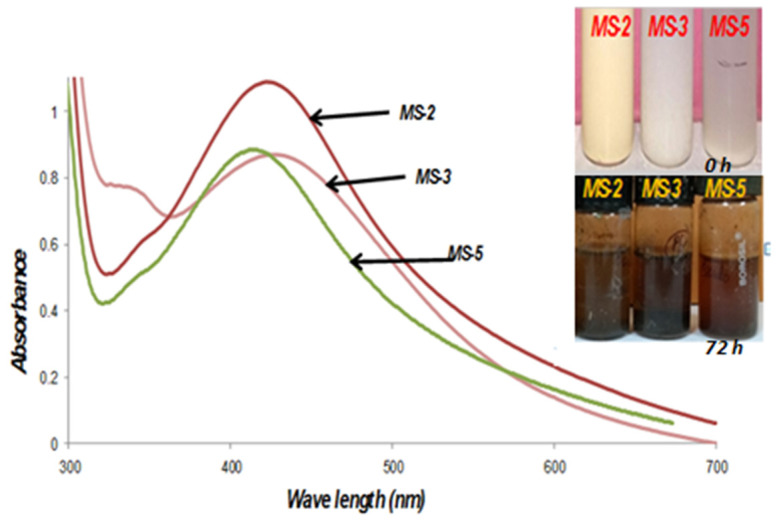
Absorbance spectra of the synthesis of silver nanoparticles at 72 h by cell pellets of endophytic bacteria isolates with absorption maxima at λmax = 415 nm.

**Figure 6 plants-11-01787-f006:**
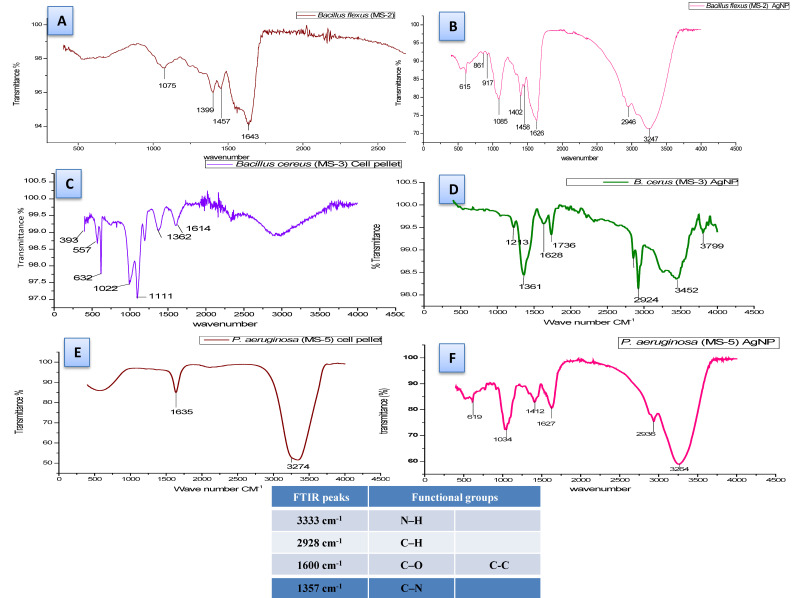
FTIR spectra of biosynthesized silver nanoparticles. (**A**) *Bacillus flexus* (MS-2) cell pellet, (**B**) *B. flexus* (MS-2) AgNP, (**C**) *Bacillus cereus* (MS-3) cell pellet, (**D**) *B. cereus* (MS-3) AgNP, (**E**) *Pseudomonas aeruginosa* (MS-5) cell pellet, and (**F**) *P. aeruginosa* (MS-5) AgNP.

**Figure 7 plants-11-01787-f007:**
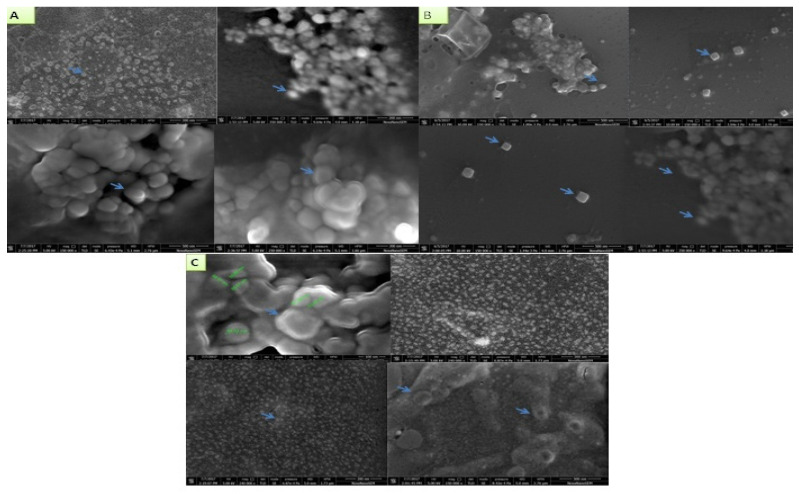
HR- SEM images of silver nanoparticles synthesized by endophytic bacterial strains (**A**) *B. flexus* (MS-2), (**B**) *B. cereus* (MS-3), and (**C**) *P. aeruginosa* (MS-5).

**Figure 8 plants-11-01787-f008:**
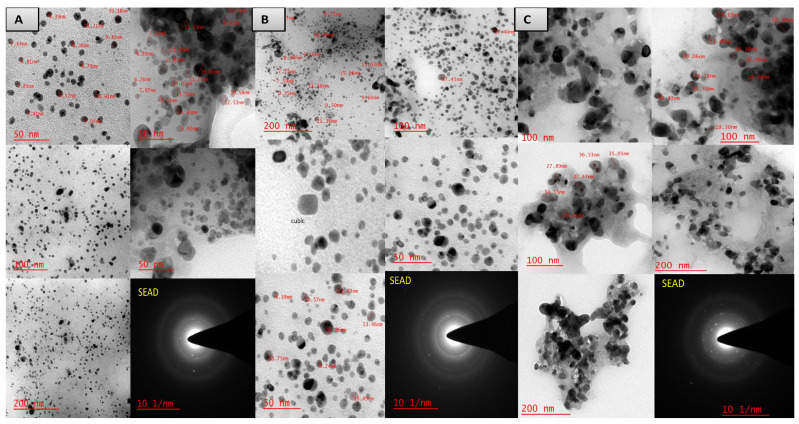
TEM images of biosynthesized AgNPs mediated by (**A**) *B. flexus* (MS-2), (**B**) *B. cereus* (MS-3), and (**C**) *P. aeruginosa* (MS-5). The TEM-SEAD ring patterns show (face-cubic centered) the crystalline nature of the AgNPs.

**Figure 9 plants-11-01787-f009:**
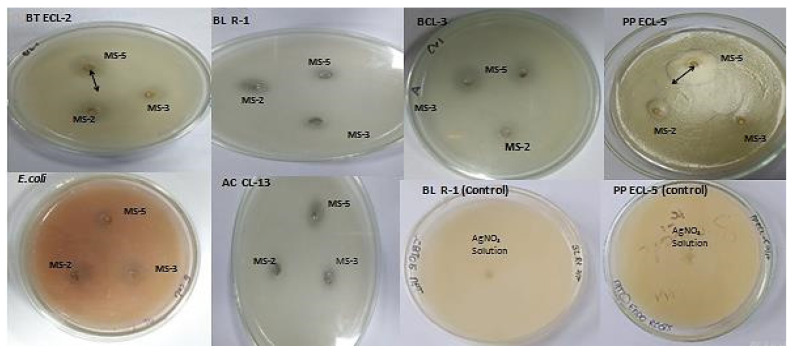
Zone of inhibition formed by endophytic bacterial strains *B. flexus* (MS-2), *B. cereus* (MS-3), and *P. aeruginosa* (MS-5) synthesized AgNPs (100 μL) on the lawns of *Bacillus thuringiensis* (BT ECL-2), *Bacillus licheniformis* (BLR-1), *Bacillus* sp., CL-3 (BCL-3), *Pseudomonas putida* (PP ECL-5), *Escherichia coli* (*E. coli* )and *Azotobacter chroococcum* (AC CL-13), *Bacillus licheniformis* (BLR-1) control, and *Pseudomonas putida* (PP ECL-5) control.

**Table 1 plants-11-01787-t001:** Effect of endophytic PGPB strain inoculation on plant growth after 21 days of sowing (L- Local, H- Hybrid).

Endophytic Bacteria	Root Length (cm) ± SD	Shoot Length (cm) ± SD
Control (Hybrid)	3.0 ± 0.4 ^d^	15.3 ± 2.5 ^e^
*Rhizobium pusense* MS-1 (H)	3.1 ± 0.5 ^d^	16.3 ± 3.0 ^e^
*Bacillus flexus* MS-2 (H)	4.5 ± 0.7 ^c^	19.3 ± 3.0 ^c^
*Bacillus cereus* MS-3 (H)	3.1 ± 0.8 ^d^	17.0 ± 4.0 ^d^
*Methylophilus flavus* MS-4 (H)	5.4 ± 0.5 ^a^	26.16 ± 3.5 ^a^
*Pseudomonas aeruginosa* MS-5(H)	4.9 ± 0.9 ^b^	21.83 ± 3.0 ^b^
Control (Local)	2.4 ± 0.2 ^d^	12.43 ± 1.5 ^e^
*R. pusense* MS-1 (L)	3.1 ± 0.3 ^c^	14.33 ± 2.2 ^d^
*B. flexus* MS-2 (L)	3.5 ± 0.3 ^c^	18.0 ± 3.0 ^b^
*B. cereus* MS-3 (L)	3.3 ± 0.7 ^c^	16.33 ± 2.3 ^c^
*M. flavus* MS-4 (L)	4.2 ± 0.7 ^a^	21.83 ± 3.0 ^a^
*P. aeruginosa* MS-5 (L)	3.6 ± 0.5 ^b^	20.86 ± 2.0 ^a^

Data are means of three replicates; values labeled with different letters (a, b, c, d, and e) are statistically significant at *p* < 0.05.

**Table 2 plants-11-01787-t002:** Morphological and agronomical traits of endophytic inoculated tomato plants and control after 60 days of plantation.

Variety	Treatments	Time Taken for Germination (in Days)	Percent Germination	Plant Height (cm)	Number of Fruits/Plant	Mean Weight (gm) of Fruit /Plant	Weight of Total Fruit (kg/Plant) after 1st Flowering
Hybrid	Control	5	62%	50 ± 4 ^a^	19 ± 3 ^a^	28 ± 4 ^a^	0.532 ^a^
	MS-1	5	66%	51 ± 3 ^a^	22 ± 4 ^a^	29 ± 5 ^a^	0.638 ^a^
	MS-2	5	75%	61 ± 4 ^b^	25 ± 3 ^b^	40 ± 3 ^b^	1.000 ^b^
	MS-3	4	74%	56 ± 4 ^a^	28 ± 2 ^b^	35 ± 3 ^c^	0.980 ^b^
	MS-4	3	78%	75 ± 8 ^c^	48 ± 4 ^c^	50 ± 3 ^d^	2.400 ^c^
	MS-5	3	69%	64 ± 5 ^b^	27 ± 4 ^b^	44 ± 3 ^b^	1.188 ^b^
Local	Control	3	78%	63 ± 7 ^a^	25 ± 5 ^a^	23 ± 1 ^a^	0.575 ^a^
	MS-1	2	90%	65 ± 4 ^a^	28 ± 3 ^a^	29 ± 3 ^b^	0.812 ^b^
	MS-2	3	85%	70 ± 6 ^a^	38 ± 2 ^b^	34 ± 3 ^b^	1.292 ^c^
	MS-3	3	96%	68 ± 7 ^a^	34 ± 2 ^a^	31 ± 4 ^b^	1.054 ^c^
	MS-4	2	99%	75 ± 5 ^a^	40 ± 3 ^b^	44 ± 4 ^c^	1.760 ^d^
	MS-5	2	95%	73 ± 4 ^a^	36 ± 5 ^a^	40 ± 4 ^c^	1.440 ^c^

Data are means of three replicates and the values labeled with different letters (a–e) are statistically significant at *p* < 0.05.

**Table 3 plants-11-01787-t003:** Evaluation of antibacterial effects of AgNPs synthesized by endophytic bacteria on pathogenic bacteria, endophytic bacterial strains, and AgNO_3_ as a control.

(100 µL/well)	Zone of Inhibition in mm					
	*Bacillus thuringiensis*	*Pseudomonas putida*	*Azotobacter chroococcum*	*Escherichia coli*	*Bacillus licheniformis*	*Rhizobium* sp.
*B. flexus* MS-2 AgNPs	17 ± 2 ^a^	8 ± 0.5 ^a^	5 ± 0.4 ^a^	7 ± 0.6 ^a^	7 ± 0.5 ^a^	7 ± 0.7 ^a^
*B. cereus* MS-3 AgNPs	5 ± 0.4 ^b^	9 ± 0.5 ^a^	7 ± 0.6 ^a^	5 ± 0.3 ^a^	10 ± 0.8 ^a^	9 ± 0.9 ^a^
*P. aeruginosa* MS-5 AgNPs	21 ± 2 ^a^	28 ± 2 ^b^	9 ± 0.8 ^a^	6 ± 0.5 ^a^	7 ± 0.6 ^a^	11 ± 1 ^a^

Data are the mean of three replicates and the values labeled with different letters (a, b, c, d, and e) are statistically significant at *p* < 0.05.

## Data Availability

Not applicable.

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
