# Peer review of "Bioprospects of Endophytic Bacteria in Plant Growth Promotion and Ag-Nanoparticle Biosynthesis"

_plants, 2022, doi:10.3390/plants11141787_

Round 1

Reviewer 1 Report

In the present study, we have used five endophytic bacterial strains and applied on the two varieties of tomato seeds and seedlings, to evaluate their impact on morphological yields of tomato.

In Introduction I missed more information about bacteria tested in study.

In material and methods:Which variety of tomatoes were used?

From which environment were species of study isolated? How were bacterial strains identified?

How were Ag-nanoparticles isolated?Antibacterial activity is described as unsatisfactory.

Results:

Some bacterial names are without italics, for example line 216-218. After the first mentioned name of bacteria, they can be written shortly.In discussion is more information about antimicrobial activity missing.

Same novelty of the manuscript is missing.

In conclusion, the manuscript is very well described, but still has some flaws.

Author Response

Response to reviewer’s suggestion:

First of all we thanks to the learned anonymous reviewers for their constructive comments, which helps in improving the quality of our article. We have revised the article thoroughly as per reviewers suggestion and the revised text are highlighted in red                                                   

Quarry In the present study, we have used five endophytic bacterial strains and applied on the two varieties of tomato seeds and seedlings, to evaluate their impact on morphological yields of tomato. In Introduction I missed more information about bacteria tested in study.

Response:  Thanks for the comments, however we have mentioned the citation as [25] in the text which briefly disused isolation and characterization of these five tested endophytic bacterial strains.  In this paper to avoid the repetition, we have not mentioned these things in the introduction section.

Quarry: In material and methods: Which variety of tomatoes were used?

Response:  Thanks for the comments, we have used two tomatoes verities S-22 HYBRID and S-3619 Local, which were procured from the horticulture section of  I.I.V.R , Varanasi, India.

Quarry: From which environment were species of study isolated? How were bacterial strains identified?

Response: The endophytic bacterial strains were isolated from the roots of Solanum lycopersicum L. surviving at 45-470C in the month of June 2013.. On the basis of morphological, biochemical, carbon utilization pattern by Biolog and 16SrRNA gene sequence analysis as mentioned in citation [25] Singh, M.; Kumar A.; Pandey, K.D. Biochemical and molecular identification of Solanum lycopersicum L. temperature tolerant bacterial endophytes. Biocat. Agric. Biotechnol. 2019. 22, 101409.

Quarry How were Ag-nanoparticles isolated? Antibacterial activity is described as unsatisfactory.

Response:  Thanks for the comments, however, in the article please see the section: 2.7 Synthesis of nanoparticles, we have discussed the method of Ag-nanoparticle synthesis. As per concern about the antibacterial activity, we have added a paragraph in the revised article

Results:

Quarry Some bacterial names are without italics, for example line 216-218. After the first mentioned name of bacteria, they can be written shortly. In discussion is more information about antimicrobial activity missing.

Response:   Done

Quarry: Same novelty of the manuscript is missing.

Response:  Thanks for the comments, we have revised the  article thoroughly and  claearly demarcated the finding in the conclusion section in the revised text.

Quarry : In conclusion, the manuscript is very well described, but still has some flaws.

Response:  Thanks to the learned anonymous reviewers for the suggestion, we have revised the manuscript thoroughly including material and methods, discussion and conclusion section in the revised article.

Reviewer 2 Report

In this manuscript entitled ”Bioprospects of Endophytic Bacteria in Plant Growth Promotion and Ag-Nanoparticle Biosynthesis”, five endophytic bacteria were applied to two tomato varieties, and the agronomy performance of these treated tomato varieties were evaluated. In addition, authors also used these five bacteria for biosynthesis of Ag-Nanoparticle. However, in my opinion,  evaluation of effect of the bacteria on tomato and  biosynthesis of Ag-Nanoparticle are different topic, why authors put these different topics into one manuscript?

In materials and methods section

Would you provide more background of two tomato varieties? Because I do not understand why you select these two varieties for assessing effect of the bacteria.

In figure 1, please describe which is treatment, and which is control.

Fig2-3 and Table 1-2 should display significant difference between the treatments.

3.6 antibacterial activity of nanoparticle section

The bacteria B.flexus, B.cereus, and P.areuginosa may be antagonist several plant pathogens. In your experiment, AgNO3 have no effect on the plant pathogens. Therefore, antimicrobial activity of AgNPs synthesized by these bacteria may be from the bacteria B.flexus, B.cereus, and P.areuginosa. Authors should also test antimicrobial activity of the endophytic bacteria, in relative to of AgNPs synthesized by these bacteria. I think this experiment need to be improved.

Figure 9 is so blur, would you provide a clear one.

Author Response

First of all we thanks to the learned anonymous reviewers for their constructive comments, which helps in improving the quality of our article. We have revised the article thoroughly as per reviewers suggestion and the revised text are highlighted in red

Quarry In this manuscript entitled ”Bioprospects of Endophytic Bacteria in Plant Growth Promotion and Ag-Nanoparticle Biosynthesis”, five endophytic bacteria were applied to two tomato varieties, and the agronomy performance of these treated tomato varieties were evaluated. In addition, authors also used these five bacteria for biosynthesis of Ag-Nanoparticle. However, in my opinion, evaluation of effect of the bacteria on tomato and  biosynthesis of Ag-Nanoparticle are different topic, why authors put these different topics into one manuscript?

Response: Thanks to the learned anonymous reviewers for the comments. We do agree with the suggestion, this article is the combination of two experiments, because in this article we try to cover the application part of these five endophytic bacterial strains.  In the previous article, we have published the isolation and characterization of  these endophytic bacterial strains, see the citation [25] Singh, M.; Kumar A.; Pandey, K.D. Biochemical and molecular identification of Solanum lycopersicum L. temperature tolerant bacterial endophytes. Biocat. Agric. Biotechnol. 2019. 22, 101409.

In materials and methods section

Would you provide more background of two tomato varieties? Because I do not understand why you select these two varieties for assessing effect of the bacteria.

Response:  Thanks for the comments, we want to clarify the anonymous reviewer that in the previous study we  had selected 8 different  (6 hybrid and  2 local ) varieties from IARI Varanasi- for the enumeration of  culturable endophytic bacterial population in that study, results concluded that S-22 hybrid and , S-3619 have more bacterial population and highest percent of germination, due to this, we have selected  these two varieties see the citation [24] Singh, M., Singh, P.P., Patel, A.K., Singh, P.K. and Pandey, K.D., Enumeration of Culturable Endophytic Bacterial Population of Different Lycopersicum esculentum L. Varieties. 2018. Int. J. Curr. Microbiol. App. Sci7(2), pp.3344-3352.

Quarry: In figure 1, please describe which is treatment, and which is control.

Response:  Thanks for the comments, we have revised the figure -1as per suggestion.

Quarry Fig2-3 and Table 1-2 should display significant difference between the treatments.

Response:  Thanks for the comments, we have revised the Fig2-3 and Table 1-2 and added the statistics  

Quarry 3.6 antibacterial activity of nanoparticle section

The bacteria B.flexusB.cereus, and P.areuginosa may be antagonist several plant pathogens. In your experiment, AgNO3 have no effect on the plant pathogens. Therefore, antimicrobial activity of AgNPs synthesized by these bacteria may be from the bacteria B.flexusB.cereus, and P.areuginosa.

Response: Thanks for the comments, we do agree with the reviewer comments and suggestions. The antibacterial activity may be due to the tested bacterial strains. In this paper we are reporting our primarily report. We are working on the section and we will highlight suggestion of anonymous reviewers in the next paper or ongoing experiments.

Quarry Authors should also test antimicrobial activity of the endophytic bacteria, in relative to of AgNPs synthesized by these bacteria. I think this experiment need to be improved.\

Response:  : Thanks for the comments, we assure anonymous reviewers, we will design some experiments to evaluate the  comparative study  of nanoparticles and endophytic bacterial strains

Quarry Figure 9 is so blur, would you provide a clear one.

Response:  Thanks for the comments, But we are very sorry, because we don’t have quality pictures of this experiment. If reviewers suggest, we can include this figure -9 in the supplementary data.

Round 2

Reviewer 1 Report

Authors accepted all comments.

Tables are not by instruction for authors.

Author Response

Quarry: 

Tables are not by instruction for authors.

Response: Thanks for this critical suggestion, we have improved and uniformed both the tables

Reviewer 2 Report

no suggestions

Author Response

Thanks a lot